# NU-MCC: Multiview Compressive Coding with Neighborhood Decoder and Repulsive UDF

**Stefan Lionar**[1,2]  **Xiangyu Xu**[3✉]  **Min Lin**[1]  **Gim Hee Lee**[2]

[1]Sea AI Lab  [2]National University of Singapore  [3]Xi'an Jiaotong University

Project page: https://numcc.github.io/

## Abstract

Remarkable progress has been made in 3D reconstruction from single-view RGB-D inputs. MCC is the current state-of-the-art method in this field, which achieves unprecedented success by combining vision Transformers with large-scale training. However, we identified two key limitations of MCC: 1) The Transformer decoder is inefficient in handling large number of query points; 2) The 3D representation struggles to recover high-fidelity details. In this paper, we propose a new approach called NU-MCC that addresses these limitations. NU-MCC includes two key innovations: a **N**eighborhood decoder and a Repulsive **U**nsigned Distance Function (Repulsive UDF). First, our Neighborhood decoder introduces center points as an efficient proxy of input visual features, allowing each query point to only attend to a small neighborhood. This design not only results in much faster inference speed but also enables the exploitation of finer-scale visual features for improved recovery of 3D textures. Second, our Repulsive UDF is a novel alternative to the occupancy field used in MCC, significantly improving the quality of 3D object reconstruction. Compared to standard UDFs that suffer from holes in results, our proposed Repulsive UDF can achieve more complete surface reconstruction. Experimental results demonstrate that NU-MCC is able to learn a strong 3D representation, significantly advancing the state of the art in single-view 3D reconstruction. Particularly, it outperforms MCC by 9.7% in terms of the F1-score on the CO3D-v2 dataset with more than $5\times$ faster running speed.

## 1 Introduction

3D reconstruction from a single-view RGBD input is a fundamental problem in computer vision with applications in robotics [1, 2] and VR/AR [3]. The state-of-the-art approach for this task is MCC [4], which leverages large-scale multi-view images [5] to develop a scalable model for 3D reconstruction from a single RGB-D image. MCC utilizes the depths and 3D point clouds for supervision obtained using the COLMAP framework [6, 7]. By combining Vision Transformer [8, 9] with large-scale training, MCC can learn a generalizable textured single-view 3D reconstruction model that generalizes to diverse zero-shot settings.

However, we have identified two key limitations of MCC that affect its reconstruction quality and model efficiency. First, the Transformer decoder in MCC directly takes in the 3D locations of query points to predict their respective occupancy and color. Due to the quadratic complexity of Transformers, this approach incurs a high computational cost when the number of query points is large, as is often the case for detailed 3D reconstruction. Second, MCC uses the occupancy field as the underlying 3D representation, which hinders the recovery of high-fidelity geometry and texture details. This can be observed in Figure 1, where the reconstruction lacks intricate details.

---

✉Corresponding Author. Work was partially done at Sea AI Lab.

37th Conference on Neural Information Processing Systems (NeurIPS 2023).

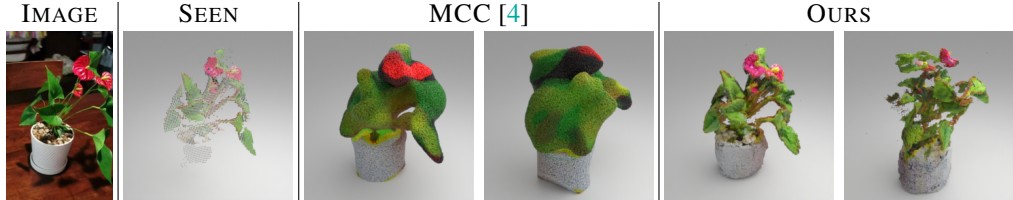

| IMAGE | SEEN | MCC [4] | OURS |
|---|---|---|---|

Figure 1: **Single-view reconstruction given RGB-D input.** Our approach achieves higher texture and geometry fidelity compared to the state-of-the-art method [4].

To address these limitations, we propose a Neighborhood decoder, where each query point only attends to a small set of features around its neighborhood and thus enhances the efficiency. One key component of the Neighborhood decoder is the anchor predictor that predicts sparse-yet-complete anchor proxies that captures the coarse shape and appearance of the input object. The Neighborhood decoder further aggregates the neighboring proxies of the query point for the final prediction. Importantly, the number of anchors is fixed and therefore the computational cost in the Transformer does not depend on the number of query points. This design not only results in a much faster inference speed but also enables the exploitation of finer-scale visual features without significantly increasing the computational cost.

Our second contribution lies in the improved design of the output 3D representation. In MCC [4], the occupancy field requires selecting a radius around ground truth points to determine if a query point is occupied or unoccupied during training. However, this radius selection poses challenges, with a small radius leading to incomplete surface coverage and a large radius causing over-thickening reconstruction. In addition, the occupancy-based surface reconstruction process is wasteful since the empty query points are discarded. To overcome these limitations, we leverage UDF, which shifts query points to the closest surface using its gradient field. UDF does not require a radius and obtains supervision signals by calculating distances to the nearest ground truth points. While iteratively shifting points can move them to the surface, our UDF analysis as detailed in Section 3.3 shows that the point-shifting process tends to move points toward high-curvature regions like corners or edges, which eventually results in hole artifacts. Therefore, we propose Repulsive UDF as an alternative to vanilla UDF that incorporates repulsive forces between nearby points. Our Repulsive UDF mitigates the hole artifacts and produces a uniform distribution of points on the surface.

In summary, our contributions are as follows:

- We propose **NU-MCC**, a new single-view 3D reconstruction model with two key innovations: 1) a Neighborhood decoder that efficiently handles large amounts of query points and combines both coarse and fine-scale visual features for 3D reconstruction, and 2) a Repulsive UDF that achieves significantly higher surface quality than the occupancy field and standard UDF.
- We conduct experiments on CO3D-v2 dataset [5] and show that our model outperforms the state-of-the-art [4] by 9.7% with more than $5\times$ faster inference speed. We also show that our method is generalizable to diverse challenging zero-shot single-view reconstruction settings, such as from iPhone RGB-D capture, AI-generated images [10], and ImageNet [11].

## 2 Related works

**Single-view 3D reconstruction.** Inferring the complete 3D shape from a single image is a challenging problem due to the ambiguity caused by occlusions. Previous works have shown impressive single-view 3D reconstruction results using voxels [12–16], point clouds [17, 18], meshes [19–21], and implicit representations [22–25]. However, the results are commonly demonstrated on simplistic synthetic dataset, such as ShapeNet [26] and Pix3D [27]. The notable exception is MCC [4] which learns compressive 3D appearance and geometry from large-scale multi-view images. While MCC [4] can generalize to diverse in-the-wild settings, it operates slowly and produces reconstruction with lacking details due to simplistic Transformer decoder that attends to all visual features. In contrast, we propose an efficient Neighborhood decoder that attends to a small set of relevant local features.

**Shape completion.** Shape completion methods aim to complete the full geometry given incomplete parts. Relevant to our approach is PoinTr [28] which predicts proxies of the complete shape to

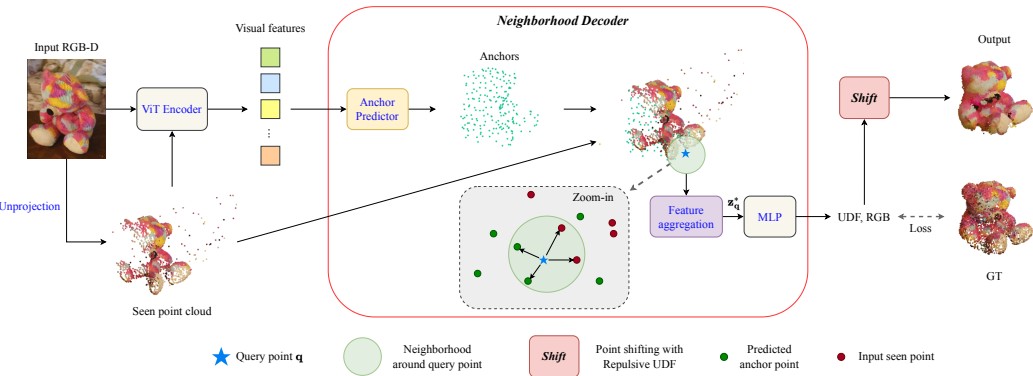

Figure 2: **Overview of the proposed NU-MCC.** Given an input single-view RGB-D image, we first unproject the pixels into the 3D world frame, resulting in a textured partial point cloud. We then employ a standard ViT to extract visual features from the partial point cloud. Next, we introduce our Neighborhood decoder, which utilizes the extracted visual features to estimate the UDF and RGB values of each query point in 3D space. Unlike existing Transformer decoders where each query point needs to attend to all the visual features, the Neighborhood decoder allows each query point (blue star in the figure) to attend to only a small set of features in its neighborhood, significantly improving the efficiency of our model. During inference, the predicted UDF is used to shift the query points to the surface of the 3D object, leading to high-quality 3D reconstruction. The anchor predictor is elaborated in Figure 3. More details regarding the point-shifting mechanism with the proposed Repulsive UDF are illustrated in Figure 4.

generate a fixed resolution of point clouds in explicit manner. Other methods employ conditional generation from learned shape priors [29, 30] for better and diverse completions, but suffer from slow generation speed and may not scale well for large-scale training data. In contrast to those methods, our approach generates implicit functions that allow any desired output resolution, and leverages large-scale data for training. Additionally, our method learns the object texture, which is not addressed by those previous methods.

**Neural fields for 3D reconstruction.** Recently, numerous methods using neural fields have been proven effective for 3D reconstruction [31]. These approaches employ various output representations, such as occupancy [32], signed distance function [33], UDF [34], and radiance fields [35]. Among these representations, UDF methods have demonstrated their effectiveness in reconstructing open surfaces [34, 36, 37]. While early works in neural fields consider simple structures of latent representations, more sophisticated structures such as planes [38, 39], grids [38, 40–42], gaussians [43, 44], and anchors [45] have been proposed for more efficient reconstruction. We adopt anchor structure for efficiency in the attention mechanism [8] in our Neighborhood decoder and propose Repulsive UDF as an alternative output representation.

## 3 Method

We introduce NU-MCC, a novel approach for 3D reconstruction from a single-view RGBD image. Inspired by MCC [4], the proposed algorithm employs an encoder-decoder architecture as illustrated in Figure 2. While following MCC to design the encoder, we propose a new Neighborhood decoder which offers improved efficiency over the baseline decoder without compromising performance. Moreover, unlike the original MCC that uses occupancy fields, we propose a Repulsive UDF that significantly enhances the reconstruction results.

### 3.1 Encoder

We use the same encoder as MCC [4] to process the input RGB-D image. Denoting $I \in \mathbb{R}^{H \times W \times 3}$ as the RGB image and $D \in \mathbb{R}^{H \times W}$ as the depth map, we first unproject the depth map $D$ into the 3D world frame and obtain the 3D positions of the pixels $P \in \mathbb{R}^{H \times W \times 3}$. Essentially, the image $I$ and 3D position map $P$ describe the textured seen point cloud, which is both noisy and incomplete as shown in Figure 2. We then embed the partial point cloud into a sequence of feature tokens $\mathbf{R} \in \mathbb{R}^{N \times d}$

using ViTs [8, 9], where $N = 14 \times 14 + 1$ is the number of tokens ($14 \times 14$ patch tokens plus a global token), and $d$ is the number of feature channels. We refer to MCC [4] for more details of the encoder.

## 3.2 Neighborhood decoder

In this section, we discuss the decoder component, which takes the output of the encoder $\mathbf{R}$ and $N_q$ point queries in 3D space to predict UDF and colors for each point.

Existing decoders [4] commonly employ a simple concatenation strategy, where the $N$ visual tokens in $\mathbf{R}$ and the $N_q$ query tokens are first concatenated and fed into normal Transformer layers. However, this approach incurs a high computational cost of $\mathcal{O}\left((N + N_q)^2\right)$, particularly when the number of queries $N_q$ is large, as is often the case for detailed 3D reconstruction.

To address this issue, our key insight is that the inefficiency stems from each query having to interact with all visual features. A better solution should allow each query to efficiently attend to a small subset of tokens that are most relevant to it. To this end, we introduce anchors as a proxy representation of visual features $\mathbf{R}$, which facilitates the identification of relevant features for each query.

**Anchor predictor.** Our anchor representation is denoted as $\mathbf{F}_c = \{\mathbf{Z}_c, \mathbf{X}_c\} \in \mathbb{R}^{M \times (d+3)}$, which consists of two parts. First, $\mathbf{Z}_c \in \mathbb{R}^{M \times d}$ is a set of anchor features, which represents transformed visual features. Second, $\mathbf{X}_c \in \mathbb{R}^{M \times 3}$ denotes the 3D locations of the anchors (green points in Figure 2), which serve as the identifier of each feature vector. $M$ represents the number of anchors, while $d$ is the dimension of each anchor vector. This design enables each query point $\mathbf{q} \in \mathbb{R}^3$ to conveniently identify its relevant features by locating the $m$-nearest anchors within its 3D neighborhood based on the Euclidean distance from $\mathbf{q}$ to $\mathbf{X}_c$.

As shown in Figure 3, we employ a Transformer $\mathcal{G}$ as the anchor predictor:

$$\mathbf{F}_c = \{\mathbf{Z}_c, \mathbf{X}_c\} = \mathcal{G}(\mathbf{R}).$$

To bootstrap the anchor predictor, we begin by randomly initializing a set of learnable positional embeddings for each anchor, denoted as $\mathbf{E}_0 \in \mathbb{R}^{M \times d}$. Subsequently, we incorporate the global token $\mathbf{z}_0$ into $\mathbf{E}_0$ through broadcasting and addition, resulting in the anchor embeddings $\mathbf{E} \in \mathbb{R}^{M \times d}$. Next, we concatenate the visual feature $\mathbf{R}$ with the anchor embeddings $\mathbf{E}$, which are then passed through standard Transformer layers, yielding the anchors $\mathbf{F}_c$ and an updated global feature vector $\mathbf{z}$. The output patch tokens (the grey squares in Figure 3) are not used in

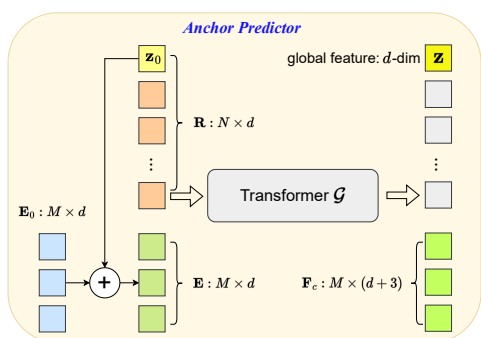

Figure 3: Illustration of the anchor predictor.

subsequent modules. Note that the computational cost of the anchor predictor is $\mathcal{O}\left((N + M)^2\right)$, which does not depend on the number of query points, making it manageable for a reasonable number of anchors $M$.

To ensure the generation of meaningful anchor representations, we directly supervise the anchor locations $\mathbf{X}_c$ during training by encouraging the anchors to be close to the ground-truth 3D surface. The anchor features $\mathbf{Z}_c$ are not directly supervised and are only trained with the final RGB and UDF prediction loss. The predicted anchors can be seen as a sparse-yet-complete point cloud that captures the coarse shape and apperance of the input object. This approach allows us to efficiently generate informative and context-aware anchor representations for our Neighborhood decoder.

**Feature aggregation.** With the proposed anchor representation, we generate the feature of each query point $\mathbf{q}$, denoted as $\mathbf{z}_{\mathbf{q}}^* \in \mathbb{R}^d$, by aggregating the anchor features in its neighborhood. Our aggregation process is derived from [46, 45] and detailed as follows:

$$\mathbf{z}_{\mathbf{q}}^* = \sum_{i=1}^{m} \mathbf{W}^i \odot (\mathbf{Z}_{c,\mathbf{q}}^i W_v), \tag{1}$$

where $\mathbf{Z}_{c,\mathbf{q}} \in \mathbb{R}^{m \times d}$ represents the $m$-nearest anchors to $\mathbf{q}$, with $\mathbf{Z}_{c,\mathbf{q}}^i$ denoting the $i$-th row. A linear projection layer $W_v \in \mathbb{R}^{d \times d}$ is applied to each anchor feature before aggregation. $\mathbf{W}^i \in \mathbb{R}^d$ denotes

the aggregation weight of the $i$-th nearest anchor $\mathbf{Z}_{c,\mathbf{q}}^i$, and $\odot$ is the Hadamard product of two vectors. The vector-based weighted sum provides more flexibility than a scalar weighted sum by allowing different weights for different channels.

The aggregation weights $\mathbf{W} \in \mathbb{R}^{m \times d}$ are estimated based on three factors: 1) the displacement between $\mathbf{q}$ and the $m$ neighboring anchors, denoted by $\Delta_{\mathbf{q}} \in \mathbb{R}^{m \times 3}$, 2) the features of the neighboring anchors $\mathbf{Z}_{c,\mathbf{q}}$, and 3) the global token $\mathbf{z}$ obtained from the anchor predictor. The estimation process can be written as:

$$\mathbf{W} = \sigma \left( \psi \left( \mathbf{z} W_q + \mathbf{Z}_{c,\mathbf{q}} W_k + \delta(\Delta_{\mathbf{q}}) \right) \right), \tag{2}$$

where both $\psi$ and $\delta$ are two-layer MLPs, and $W_q$ and $W_k$ denote linear projection layers. $\sigma$ is a softmax function along the first dimension, normalizing the aggregation weights across different anchors. The global token vector $\mathbf{z}$ is broadcasted before being added to the matrices.

Finally, we can map the aggregated feature $\mathbf{z}_{\mathbf{q}}^*$ to the UDF $f(\mathbf{q})$ and color $c(\mathbf{q})$ of the query point $\mathbf{q}$:

$$f(\mathbf{q}), c(\mathbf{q}) = \Psi(\mathbf{q}, \mathbf{z}_{\mathbf{q}}^*) \tag{3}$$

where $\Psi$ is an MLP as shown in Figure 2.

**Incorporating fine features.** Considering the large patch size used in the ViT encoder, typically $16 \times 16$, we hypothesize that the encoded representation $\mathbf{R}$, as well as the anchors features $\mathbf{Z}_c$ obtained from $\mathbf{R}$, mainly capture coarse information from the input. This limitation hinders existing methods, such as MCC, from recovering fine details in the reconstructed output.

In addition to the high efficiency, another advantage of our proposed Neighborhood decoder is its ability to conveniently incorporate fine-scale information directly from the input, without significantly increasing computational costs. Specifically, as the RGB-D input provides a textured partial point cloud as introduced in Section 3.1, we can transform the input into a format similar to the anchors and construct fine features as $\mathbf{F}_f = \{\mathbf{Z}_f, P\} \in \mathbb{R}^{(HW) \times (d+3)}$. Here, $\mathbf{Z}_f \in \mathbb{R}^{(HW) \times d}$ is obtained by applying a linear projection layer to the RGB values, representing the low-level color information of each pixel. $P$ represents their corresponding locations in 3D space as introduced in Section 3.1.

To integrate the coarse anchor features $\mathbf{F}_c$ with the fine input features $\mathbf{F}_f$, we identify the $m$ nearest features from $\mathbf{F}_c$ and the $n$ nearest features from $\mathbf{F}_f$ for each query point $\mathbf{q}$. These features are concatenated to form a new feature set $\mathbf{Z}_{\mathbf{q}} \in \mathbb{R}^{(m+n) \times d}$, accompanied by their respective 3D locations. Subsequently, we can perform the feature aggregation as in Eq. 1 by replacing $\mathbf{Z}_{c,\mathbf{q}}$ with $\mathbf{Z}_{\mathbf{q}}$. By aggregating the concatenated feature set $\mathbf{Z}_{\mathbf{q}}$, we achieve an enhanced representation that incorporates both coarse and fine-scale information, facilitating more detailed and accurate 3D reconstruction.

**Model flexibility.** Since the query points in our Neighborhood decoder attend to features pinpointed in continuous space, our Neighborhood decoder provides some flexibility to control the reconstruction quality. First, a higher resolution of fine input features can be adopted at test time in the feature aggregation process to generate higher-detailed reconstruction. Second, the feature aggregation mechanism in our Neighborhood decoder allows a flexible number of features that can differ during training. Aggregating higher number of features result in cleaner and smoother reconstruction that can be helpful in handling input with noisy depth.

### 3.3 Repulsive UDF

MCC [4] utilizes the occupancy field to represent 3D objects. While achieving state-of-the-art performance, the occupancy representation faces two significant challenges. First, during training, determining whether a point is occupied involves checking if it falls within a radius of any point in the ground-truth point cloud. This causes the issue of selecting an appropriate radius. A small radius may lead to incomplete surface coverage, resulting in false supervision signals. Conversely, a large radius can cause over-thickening of reconstructed results, as many non-surface points may be classified as occupied.

Second, during inference, we need to uniformly query the 3D space for reconstruction. Thus, a significant portion of the query points will be empty and do not directly contribute to the reconstructed surface. This could lead to insufficient density in the 3D reconstruction, as many areas on the surface may be under-queried and lack an adequate number of points for accurate reconstruction.

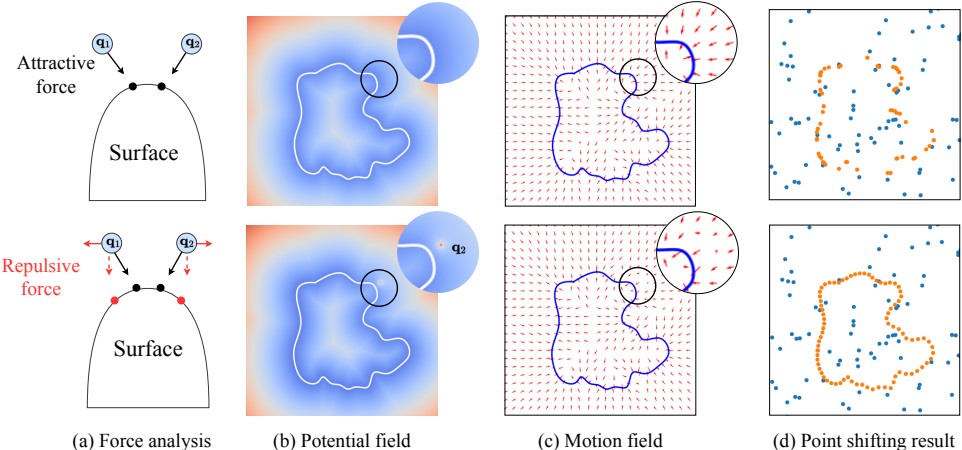

|  |  |  |  |
|---|---|---|---|
| (a) Force analysis | (b) Potential field | (c) Motion field | (d) Point shifting result |

Figure 4: **Comparison between the standard UDF (top) and the Repulsive UDF (bottom).** By improving the potential field (b), we rectify the motion field (c) for the point-shifting process, resulting in improved surface reconstruction in (d). The blue points in (d) are the initial query points randomly sampled in the space, and the orange ones are the queries after point shifting.

Recent studies [34] show that UDF addresses the above issues effectively. The UDF $f(\mathbf{q})$ is defined as the closest distance from the point $\mathbf{q}$ to the surface. During the training of UDF, one can directly regress the distance to the closest point in the ground truth, avoiding the radius selection dilemma in the occupancy field.

Meanwhile, as $f(\mathbf{q})$ is a potential field as shown in Figure 4(b)-top, the points in the 3D space can move to the surface by flowing from a high-potential region to a low-potential region, resembling the water or electricity flow. Thus, during inference, most of the queried points can be iteratively shifted to the surface with the following process:

$$\mathbf{q} \leftarrow \mathbf{q} - f(\mathbf{q}) \cdot \frac{\nabla_{\mathbf{q}} f(\mathbf{q})}{\|\nabla_{\mathbf{q}} f(\mathbf{q})\|}, \tag{4}$$

where the point $\mathbf{q}$ moves along the gradient field $\nabla_{\mathbf{q}} f(\mathbf{q})$ in Figure 4(c)-top, which is induced from the potential field. $\|\cdot\|$ is the Euclidean norm.

In this work, we introduce UDF into NU-MCC, which ensures a greater number of query points contribute to the formation of the 3D surface. However, we find that the standard UDF suffers from a drawback that the query points tend to be attracted to regions with large curvature during point shifting, which often leads to holes in flat regions. We visualize this observation with a toy example in Figure 4(a)-top, where the two query points $\mathbf{q}_1$ and $\mathbf{q}_2$ are drawn towards the corner area of the surface with high curvature. This phenomenon is also evident in the motion field in Figure 4(c)-top, where a large portion of arrows point to the corner region. As a result, the reconstructed surface exhibits undesirable holes as shown in Figure 4(d)-top.

To mitigate this issue, we propose the Repulsive UDF to encourage a more uniform distribution of query points on the reconstructed surface. As shown in Figure 4(a)-bottom, we aim to develop a new potential field that provides repulsive force between $\mathbf{q}_1$ and $\mathbf{q}_2$, preventing an excessive concentration of points in certain areas. Thus, we design the new potential field for $\mathbf{q}_1$ as:

$$f(\mathbf{q}_1) + \lambda g(\|\mathbf{q}_1 - \mathbf{q}_2\|), \tag{5}$$

where $g(x)$ is a monotonically decreasing function that assigns a high potential when $\mathbf{q}_1$ moves too close to $\mathbf{q}_2$. We empirically find $g(x) = -\ln(x)$ yields stable results. The rectified potential field and vector field are shown in Figure 4(b)-bottom and (c)-bottom. To shift to the low-potential region of Eq. 5, we alternatively move in the inverse direction of $\nabla_{\mathbf{q}_1} f(\mathbf{q}_1)$ and $\nabla_{\mathbf{q}_1} g(\|\mathbf{q}_1 - \mathbf{q_2}\|)$, fostering a more balanced transition to the 3D surface.

Additionally, to incorporate more surrounding points to build the potential function, we select $k$-nearest points of a query $\mathbf{q}$, denoted as $\mathcal{N}_{\mathbf{q}}$. This leads to the gradient field:

$$\nabla_{\mathbf{q}} \sum_{i \in \mathcal{N}_{\mathbf{q}}} g(\|\mathbf{q} - \mathbf{q}_i\|) = \nabla_{\mathbf{q}} \sum_{i \in \mathcal{N}_{\mathbf{q}}} \ln(\frac{1}{\|\mathbf{q} - \mathbf{q}_i\|}) = \sum_{i \in \mathcal{N}_{\mathbf{q}}} \frac{\mathbf{q} - \mathbf{q}_i}{\|\mathbf{q} - \mathbf{q}_i\|^2}. \tag{6}$$

To prevent excessively large forces, we apply gradient clamping during implementation. As illustrated in Figure 4(d)-bottom, the proposed Repulsive UDF effectively addresses the problem of UDF and achieves a more comprehensive surface reconstruction.

## 3.4 Optimization details

We supervise the coarse anchor locations $\mathbf{X}_c$ using $L_1$-chamfer distance loss with respect to the ground truth points sampled sparsely using farthest point sampling (FPS). The UDF predictions are supervised by the UDF between query and ground truth points using $L_1$ loss with clamping of maximal distances [34]. Following MCC [4], the colors are supervised by the colors of closest ground truth points within 0.1 radius from the query points with cross-entropy loss for each color channel. We randomly sample 550 query points from the 3D world space per training batch.

## 4 Experiments

We conduct extensive experiments to show the representational power and generalization capability of NU-MCC for object-level single-view reconstruction using CO3D-v2 dataset [5]. Additionally, we show the zero-shot generalization for in-the-wild reconstruction of RGB-D iPhone capture, AI generated images [10], and ImageNet [11] in Section 4.2.

**Dataset.** We follow the task established from MCC [4] which uses CO3D-v2 dataset [5] for single-view object reconstruction. CO3D-v2 contains approximately 37,000 short videos of 51 object categories in real-world settings. We use the training-validation split from MCC all categories experiment. The reconstruction task focuses on the foreground objects identified by the segmentation masks provided in CO3D dataset. CO3D provides the point cloud of the object and depth maps obtained from the videos via COLMAP [6, 7], albeit noisy. The sparsely sampled point clouds (20,000 points) from the COLMAP reconstruction serve as ground truths. The objects are normalized to have zero-mean and unit-variance. The query points are sampled from $[-3, 3]$ along each axis.

**Metrics.** To evaluate the geometry, we report the F1-score with distance threshold of $0.1$ following MCC [4]. Additionally, we also show the $L_1$-chamfer distance (CD) for evaluations omitting the sensitivity of the chosen distance threshold in F1-score. We also introduce $L_1$-RGB distance metrics to evaluate the colors. The $L_1$-RGB distance calculates the $L_1$-norms of the RGB difference between the predicted points and their nearest ground truths located within 0.1 radius. We detail the mathematical formulation of the $L_1$-RGB distance in the supplementary material.

**Training details.** Our model is trained with an effective batch size of 512 using $4\times$NVIDIA A100 GPUs for 100 epochs. One epoch takes approximately 2 hours. We follow the optimizer and 3D data augmentation of MCC. Adam optimizer [47] with base learning rate of $10^{-4}$, cosine schedule, and linear warm-up for the first $5\%$ of iterations are used. 3D data augmentation is performed by random scaling of $s \in [0.8, 1.2]$ and rotation $\theta \in [-180°, 180°]$ along each axis.

**Implementation details.** We employ 200 coarse anchor representations. Each query point attends to its 4-nearest coarse anchor features and 4-nearest fine features in the feature aggregation. During training, we set the distance clamping for UDF supervision to 0.5. At test time, we shift points with an initial UDF prediction lower than 0.23 for 10 iterations. To calculate the repulsive force, we process batches of 48,000 points where each point considers $k = 16$ nearest neighbors. We clamp the shifting gradient to $[-0.03, 0.03]$ in each direction.

### 4.1 Results on CO3D-v2 Validation Set

The quantitative results on CO3D-v2 [5] validation set are summarized in Table 1. We first conduct an ablation study of our decoder design with occupancy as the output representation. The occupancy is supervised following MCC [4] where a query point is considered "occupied" if it is located within a radius of 0.1 to a ground truth point, and "unoccupied" otherwise. Utilizing our Neighborhood decoder with only coarse anchor features results in comparable metrics to MCC [4], but with $15\times$ inference speedup. The incorporation of fine features in the feature aggregation process produces notably more accurate colors and slightly better geometry.

Replacing the occupancy with UDF potentially enhances the geometry and color. However, as seen in Figure 5, iteratively shifting points towards surface results in holes due to some points clumping in

| ARCHITECTURE | REPRESENTATION | $L_1$-CD$\downarrow$ | F1$\uparrow$ | $L_1$-RGB$\downarrow$ | RUNTIME (S)$\downarrow$ |
|---|---|---|---|---|---|
| MCC [4] | Occ | 0.284 | 76.4[1] | 0.376 | 18.5 |
| Ours (no fine) | Occ | 0.292 | 76.8 | 0.374 | **1.2** |
| Ours | Occ | 0.282 | 79.0 | 0.340 | 1.5 |
| Ours | UDF | 0.264 | 78.2 | **0.316** | 3.2 |
| | RepUDF | **0.237** | **83.8** | 0.320 | 3.5 |

Table 1: **Quantitative results on CO3D-v2 [5] validation set.** Results are obtained using 216k query points and averaged over three different views.

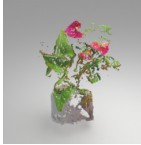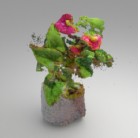

Figure 5: **The effect of repulsive force.** *Left:* Standard UDF. *Right:* Repulsive UDF.

| IMAGE | SEEN | GT | MCC [4] | OURS |
|---|---|---|---|---|

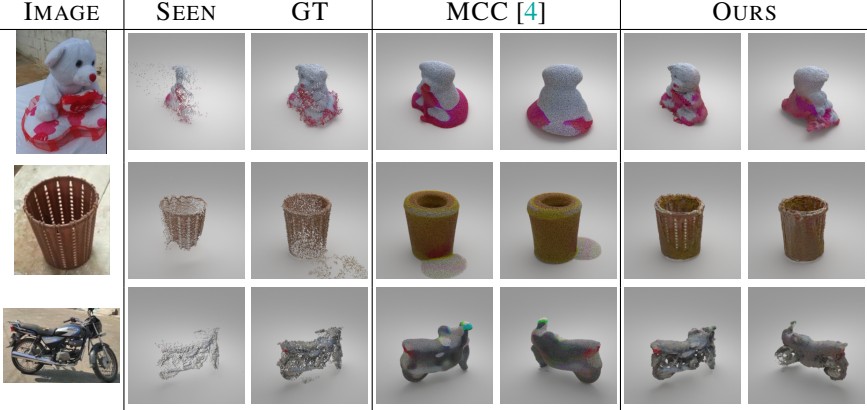

Figure 6: **Qualitative comparison on CO3D-v2 [5] validation set.** Our method captures higher details on the seen part and predicts the occlusion more accurately.

corners and edges. The use of the repulsive field mitigates the hole artifacts and creates surface with uniformly distributed points, which overall produces the best geometry and color.

Figure 6 shows the qualitative comparison with MCC [4]. While MCC [4] fails to reconstruct the fine details, such as the eyes of the teddybear and the intricate structure of the motorcycle, our method recovers significantly better details on the seen part, while more sensibly reconstructing the occlusion.

## 4.2 Zero-Shot Generalization In-the-Wild

We show the generalization capability of NU-MCC for in-the-wild reconstructions of RGB-D iPhone capture, AI generated images (Stable Diffusion [10]), and ImageNet [11]. Off-the-shelf segmentation tool [48] and depth estimator [49] are used to generate the object masks and depth images of the ImageNet [11] and AI generated [10] images. These experiments are challenging due to the large distribution shifts of depth images, object categories, and scene settings. Figure 7 shows NU-MCC is capable of accomplishing faithful reconstructions in zero-shot settings, including challenging object classes such as lawnmower in ImageNet [11] example.

## 5 Conclusion

We present NU-MCC, a novel single-view 3D reconstruction model leveraging large-scale training. Our approach includes two key innovations: a Neighborhood decoder and Repulsive UDF that in combination result in fast inference speed and high-quality textured reconstruction. We show that our model is generalizable to challenging zero-shot settings with large distribution shifts from training data, such as iPhone capture, AI generated images, and images curated from the web.

---

[1]We discovered and rectified a bug in MCC [4] F1-score calculation. Additionally, we use a score threshold of $0.4$ that results in the best chamfer distance. Thus, our reported F1-score is much higher than in MCC paper.

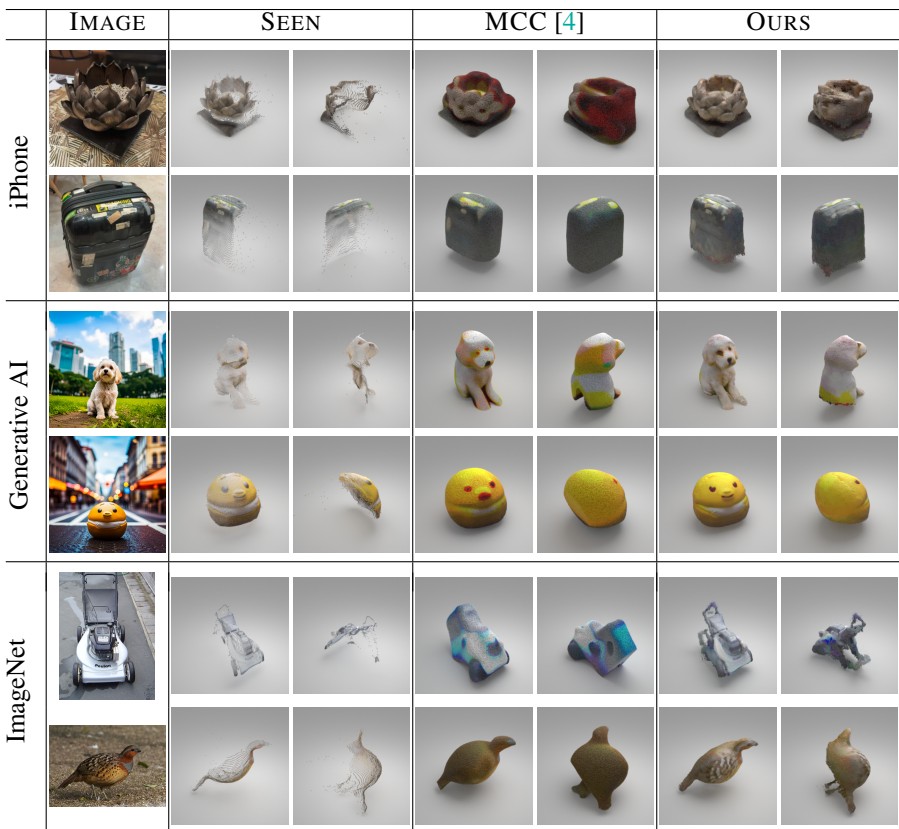

Figure 7: **Zero-shot generalization in-the-wild.** NU-MCC trained on CO3D-v2 [4] is capable of faithful reconstructions of challenging in-the-wild images with large distribution shifts of depths, object categories, and scene settings with higher details than MCC [4].

**Limitation.** While NU-MCC produces remarkable results compared to the state-of-the-art [4], it is still inherently challenging to reconstruct the occluded parts with high-fidelity details. Additionally, our results degrade when there is a significant amount of outliers and noise in the input depth, which may result in "spikes" or "bumps", although these artifacts can be alleviated by selecting a higher number of features in the feature aggregation.

## Acknowledgments

This research is supported by the National Research Foundation, Singapore under its AI Singapore Programme (AISG Award No: AISG2-RP-2021-024), and the Tier 2 grant MOE-T2EP20120-0011 from the Singapore Ministry of Education. The authors acknowledge the computational resources provided by the HPC platform of Xi'an Jiaotong University.

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
