# OpenReview forum: "NU-MCC: Multiview Compressive Coding with Neighborhood Decoder and Repulsive UDF"
_NeurIPS.cc/2023/Conference — NeurIPS 2023 poster_

### Official Review · Reviewer_AYkM · 2023-06-22

**Soundness:** 2 fair
**Presentation:** 3 good
**Contribution:** 3 good
**Rating:** 4
**Confidence:** 5

**Summary:**

The paper studies the problem of shape reconstruction from single-view RGBD images and addresses two key limitations in a major prior work, MCC: 1) inefficient decoder of quadratic time complexities, and 2) suboptimal implicit representation for pointclouds.

To improve computational efficiency, the authors propose an anchor-based neighborhood aggregation scheme to downsample the seen points. They also propose a way to sample features from the high-resolution seen points directly that improves the modeling of the fine surface details.

On the other hand, instead of using occupancy representation, UDF is adapted in this paper for higher sample efficiency and more accurate surface modeling. To further resolve the problem of point concentration, the authors add a concentration penalty in the energy function.

The authors show evaluation of their model on CO3D and HyperSim (supplement) and demonstrate state-of-the-art performance over MCC. An ablation of different designs is also included.


**Strengths:**

1. The limitation of the prior work (MCC) is well analyzed in this paper.
2. The design of the repulsive UDF makes sense and looks novel.
3. The paper is relatively easy to follow.

**Weaknesses:**

1. Missing quantitative comparison to PointTr. Based on my understanding, PointTr leverages a similar approach of anchor+neighbor aggregation. Given the results in MCC (Table 1(f)), PointTr performs significantly worse than MCC's decoder, while the finding in this paper shows similar idea to PointTr actually leads to better performance. A detailed discussion on this seemingly contradictory finding, as well as an apple-to-apple comparison to a baseline with PointTr's feature aggregation method would be very helpful.

2. It is not clear where the major speedup of the inference actually comes from. In L116, it is stated that MCC's decoder is quadratic in the number of query points. However, if my understanding is correct, this is more of an implementation/engineering issue. In fact, query points do not communicate with each other in both works, so the effective time complexity should be linear on the number of query points if implemented well. My worry here is that if the authors implement MCC in some naive quadratic way while implement their method in the effective linear way, the main speedup can actually come from this implementation difference rather than the claimed novelty. It is very important to clarify on this.

3. It would have been very helpful to break down CD into accuracy and completeness. The RepUDF representation makes sense in terms of completeness improvement. But I think it is very important to see whether it helps accuracy as well.

4. The usage of nearest points to improve fine details makes sense for seen surface, but does not make much sense for unseen surface. Is it possible to evaluate the model's performance on seen and unseen surface separately? If it only improves RGB/surface reconstruction quality on seen surface, I wonder whether a simple baseline that directly overwrite the seen part of the reconstruction with seen points would actually perform similarly to the proposed method. For example, for every queried point in the reconstruction, if there are any seen points that are close enough to this queried point, we can replace the queried point with the seen point.


**Questions:**

Please see the weakness for my concerns and questions. My final rating will be highly dependent on authors' response to these questions.

**Limitations:**

The limitations of the work have been adequately discussed.

---

> ### Author Rebuttal · Authors · 2023-08-10
>
> **Comparison to PoinTr:** There are fundamental differences between PoinTr and our NU-MCC. First, PoinTr is based on an explicit 3D representation and can only predict a fixed number of points. In contrast, our method allows an arbitrary number of output points, providing better flexibility and higher-resolution reconstruction. Besides, PoinTr employs FPS and Graph network to extract features while we use ViTs similar to MCC.
>
> Second, to generate the 3D output, we use neighborhood feature aggregation to predict repulsive UDF, while PoinTr uses FoldingNet to generate pointclouds. Note that the feature aggregation operation in PoinTr is only used to predict the center points instead of the final output pointclouds. In contrast, our anchor prediction process uses a simple Transformer structure, which does not involve any neighborhood feature aggregation.  Moreover, the aggregation strategies are also significantly different (adaptive weighted sum as in Eq. 1-2 of our paper vs max-pooling of PoinTr).
>
> Third, PointTr only predicts the missing parts of the point cloud and concatenates them with the input point cloud to obtain the complete output. Consequently, it cannot deal with noisy input point clouds that are commonly encountered in real applications. In contrast, our method predicts the complete 3D representation from a noisy partial point cloud, which is more suitable for many real-world scenarios.
> In addition, our method takes RGB-D input, while PoinTr can only handle a partial point cloud without color information. Thus, PoinTr lacks texture recovery capacity.
>
> **Speedup:** We use the official implementation of MCC instead of re-implementing it ourselves. In MCC (and also our method), the number of query points ($N_q$) is quite large, often in the order of 100k or millions. Processing them all at once in MCC’s 8-layer Transformer decoder incurs a computational complexity of O(($N_e + N_q)^2$), which is computationally prohibitive ($N_e$ is the number of encoded features). Note that in MCC, each query point needs to attend to both the input features and the query itself, making a more efficient implementation non-trivial. To circumvent this issue, MCC proposes to run the query points in batches. Suppose the batch size is $B$, MCC needs to run a for-loop with a length of $N_q/B$, processing $B$ query points in each iteration. This incurs a computational complexity of O($(N_q/B) \cdot (Ne+B)^2$) that is linearized w.r.t. the total number of query points to be processed ($N_q$).
>
> However, there is a dilemma here. A small batch size $B$ increases the first term and necessitates running a longer for-loop, while a larger $B$ leads to a quadratically growing computational complexity in the second term. We have used a number $B$ that leads to the most optimal runtime for MCC, as reported in Table 1 in our paper. It is worth mentioning that we also considered another strategy: parallelizing the $N_q/B$ runs instead of a for-loop. However, this operation is also inefficient as the encoded features ($N_e$) need to be copied $N_q/B$ times, leading to burdensome memory management overhead.
>
> In contrast, our method only has a small fixed computational complexity of O($(N_e + M)^2$) in the 8-layer Transformer decoder ($M$ corresponds to the number of predicted anchors and is a small number). The burden to process $N_q$ query points is then shifted into the feature aggregation process that is far more efficient than 8-layer Transformer and incurs a linear computational complexity of O($N_q \cdot k$), where $k$ is the number of nearest neighbors.
>
> **Breakdown into accuracy and completeness:** Please find the breakdown in Table A of the PDF. Repulsive UDF leads to marginally lower precision but significantly better recall, culminating in an overall improvement in the F1 score.
>
> **Baseline by replacing to seen points directly:** As the seen point clouds are quite noisy, directly replacing the queried point with the seen point leads to noisy reconstructions. For a better analysis, we compare with a baseline using model trained without incorporating any fine features and replacing the color of a point to the nearest seen point if it is within a small distance. As shown in Table B of the PDF, this baseline does not perform as well as the proposed method where fine features are more effectively incorporated by neighborhood aggregation. Visual examples are shown in Figure B of the PDF.

---

> ### Comment · Reviewer_AYkM · 2023-08-13
>
> I had some misunderstanding of the method earlier, and the author's response makes it clear now. I am happy to raise my rating to the positive side.

---

### Official Review · Reviewer_iFNG · 2023-06-22

**Soundness:** 3 good
**Presentation:** 3 good
**Contribution:** 4 excellent
**Rating:** 6
**Confidence:** 4

**Summary:**

This paper approaches the task of 3D reconstruction given a single RGBD image. It builds off of recent SOTA approach MCC, adding four main contributions: (1) applying attention on only anchor points instead of each query point and using weighted interpolation, (2) using linear projection for “fine” features at each input point (3) replacing occupancy representation with UDF, (4) adding a “repulsive” component to UDF. Experiments on CO3D-v2 validation set show quantitatively (1) provides significant speedup, (2) improves fidelity, (3+4) improve fidelity further. Qualitative and quantitative results show favorable performance vs. MCC.

**Strengths:**

Method’s contributions are strong:

- The quadratic nature of attention means carefully constructing queries can yield efficiency. This combined with the insight of extracting fine detail through linear projection is a nice combination.
- Repulsive UDF is intuitive and positive contribution that is clearly better than occupancy. In the well-studied field of reconstruction, this is a helpful insight.
- In the quantitative result that is reported, results are compelling. Ablations show (1) anchors provide significant speedup, (2) fine features improve fidelity, (3+4) Rep. UDF improves fidelity further. A qualitative example is convincing the repulsive UDF is meaningfully better than UDF.

Writing is generally excellent.

- For the most part, one could reproduce most of the method and experiments from reading the paper.
- Technical details are well-motivated through a thorough comparison with MCC. The reader feels like an expert in these methods after reading the paper.


**Weaknesses:**

Edit: after reading the rebuttal and other reviewer comments, the lack of comparisons weakness is remedied.

Lack of quantitative comparisons and metrics; existing evaluation is inconsistent with prior work.

- The key contributions (1-4 from summary) are validated in only one quantitative result (and one quantitative result in Supplemental). Results on CO3D are not broken down into seen and unseen categories, which would be helpful.

- Why not include more qualitative results in Supplemental? Two examples on e.g. ImageNet are not convincing, especially when no quantitative comparison is reported.

- The existing quantitative results do not follow the evaluation protocol of prior work (MCC) -- the paper appears not to report the accuracy, completeness, absolute error or MSE metrics of MCC. The numbers reported for MCC following F1-score metric do not match the prior work – presumably this is because numbers are reported on the validation split, while MCC uses the test split? This is a strange choice. Why are F1 metrics on Hypersim are significantly different from MCC?

- No comparison or at the very least reference to PointE is surprising. This relates to why I gave a 3 instead of 4 for presentation: it is strange the paper builds off MCC (unpublished, arXiv Jan 23), while ignoring entirely Point-E (unpublished, arXiv Dec 22). While not comparing to unpublished work is acceptable, it is strange to not use this work which other works (e.g. 3DFuse, Zero 123) have found outperforms MCC.


**Questions:**

To further strengthen the contributions of this paper, a more thorough ablation study upon contributions could really help. For instance:

- Does increasing the resolution of fine features help, as claimed in L187?
- How sensitive are results to the number of anchors or anchor radius?
- Do details of repulsive UDF sampling matter?

Other questions:
- I don’t follow the section on bootstrapping the anchor predictor (L131). How is global token z0 used exactly? Where does z0 come from?
- Are the same weights used for both coarse and fine, given they are concatenated and in practice the same number of coarse and fine weights are used at inference?
- Figure 3 could use a key, as currently one has to look at the text to understand int
- Figures 2-3 have text that is very small, making it a bit hard to read
- L147: “depends” should be “depend”

**Limitations:**

Yes

---

> ### Author Rebuttal · Authors · 2023-08-10
>
> Thank you for finding our proposed method compelling. We address your concerns below.
>
> **F1-score metrics do not match the prior work:** As explained in the footnote in our paper (page 8), there is a bug in MCC codebase F1 calculation. The bug can be traced on their GitHub (as of the writing of this rebuttal) → MCC/engine_mcc.py → Line 44. In the recall calculation, they loop using the number of predicted points (predicted_xyz), but then divide the summation using the number of ground truth points (gt_xyz) in Line 49. This results in a lower recall than it should be when the number of predicted points is smaller than the number of ground truth points. We have rectified this issue in our evaluation by changing predicted_xyz to gt_xyz in Line 44.
> Note that fixing the bug is in favor of the baseline method. The F1-score comparison between MCC and NU-MCC is 56.7 vs 83.8 before fixing the bug, and is 76.4 vs 83.8 after fixing the bug. This is because our method always generates dense points greater than the number of ground truths, whereas MCC with a high occupancy threshold generates fewer points. Thus, the bug penalizes MCC but not our method.
>
> **F1-score difference for Hypersim:** We believe the bug is a major reason that causes the difference. On top of it, MCC performs evaluation using the scene meshes that are not freely available and need to be purchased, while we directly use the ground truth provided by Hypersim.
>
> **CO3D seen/unseen categories:** As suggested, we show the results on CO3D following MCC’s novel categories setup (unseen categories). Due to the constrained timeframe of the rebuttal phase, we train both models for 20 epochs. As shown in the table below, the proposed method performs favorably against MCC on unseen categories.
>
> | **Model** |  **Color (&#8595;)** | **Prec (&#8593;)** | **Recall (&#8593;)** | **F1 (&#8593;)** | **Acc (&#8595;)** | **Comp (&#8595;)** | **CD (&#8595;)** |
> |-----------|-----------|----------|------------|--------|---------|----------|--------|
> | **MCC**   |     0.363 |     56.8 |       89.3 |   68.9 |   0.172 |    0.144 |  0.316 |
> | **Ours**  |     0.307 |     79.2 |       84.0 |   80.9 |   0.121 |    0.146 |  0.266 |
>
> **More metrics:** As suggested, we provide more metrics, including precision, recall, accuracy,  and completeness in Table A in the PDF.
>
> **More qualitative results:** We provide more qualitative results on the video at the link.
>
> **Comparison to Point-E:** As Point-E and our model uses different training data, it is difficult to make a fair comparison on the CO3D dataset on which our model is trained. Moreover, as Point-E is based on a fundamentally different concept (diffusion model) from our work, it is non-trivial to retrain Point-E on CO3D. Thus, we directly compare the two methods in an in-the-wild setting using images captured by iPhone. As shown in Figure C of the PDF, our model can generate denser point clouds with higher fidelity to the input than Point-E.
>
> In addition to the improved reconstruction quality, our method is also superior in terms of runtime. Note the methods, such as Point-E, 3DFuse, Zero123, involve iterative generation process and/or per-sample optimization, resulting in a much slower running speed. For example, it takes 80 seconds for Point-E to generate 4096 points, 30 minutes for 3DFuse generation, and 15 minutes for Zero123. In contrast, our model is a generalizable one-step feed-forward network, resulting in fast inference that can generate ~200k points in 3.5 seconds or faster.
>
> **Resolution of fine features:** As shown in Sec. 2.1 of the supplement, increasing the resolution of fine features typically facilitates reconstruction of more intricate details. Meanwhile, this will increase the computation cost (see response to Reviewer WpNt - Input resolution).
>
> **Number and radius of anchors:** For the effect of the number of anchors, see the response to Reviewer WpNt - Analysis of the effect of different anchor point numbers. For the radius, increasing the anchor radius (including more points in aggregation) gives a smoothing effect as shown in Sec. 2.2 of the supplement.
>
> **Details of Repulsive UDF sampling:** On Line231-232, we select k-nearest points of a query point to construct the repulsive potential field. While we use a default k=16, the Repulsive UDF is robust to different choices with stable performance on k=4, 8, 32, 64.
>
> **Global token z0:** As introduced on Line108, the global token $z_0$ is obtained from the encoder ViTs. It is the global token of the encoded features $\mathbf{R}$.
>
> **Are the same weights used for coarse and fine features?** Yes
>
> **Figures and typo:** We will rectify accordingly

---

> > ### Comment · Reviewer_iFNG · 2023-08-15
> > **Reviewer Response to Author Rebuttal**
> >
> > After reading the rebuttal and other reviews I raise my rating to 6 - weak accept.
> >
> > I feel this paper should be accepted because the contributions of the method are strong, and comparison to prior work is thorough. My main concerns were surrounding evaluation details. These, e.g. F1 score or additional comparisons, were clarified or addressed in the rebuttal.

---

### Official Review · Reviewer_WpNt · 2023-06-25

**Soundness:** 3 good
**Presentation:** 3 good
**Contribution:** 3 good
**Rating:** 7
**Confidence:** 4

**Summary:**

The paper introduces a novel method, NU-MCC, for reconstructing 3D objects from single-view RGB-D images. The proposed approach builds upon the existing MCC method by incorporating two important advancements. Firstly, it introduces a neighborhood decoder to efficiently manage a large number of query points, which helps to reduce computational complexity and improve processing speed. Secondly, it adopts a Repulsive unsigned distance function (Repulsive UDF) to enhance the quality of surface reconstruction, resulting in more accurate and detailed reconstructions. The authors have conducted extensive experiments on multiple datasets to evaluate the effectiveness of their proposed approach. The results demonstrate that NU-MCC outperforms the existing state-of-the-art methods, achieving higher accuracy and better quality reconstructions.


**Strengths:**

This paper identifies and addresses two major limitations of the state-of-the-art MCC method. The first limitation is related to the Transformer decoder, which struggles to handle a large number of query points efficiently. The second limitation is associated with the inability of the 3D representation to recover high-fidelity details accurately.

To address these limitations, the authors propose a new method called NU-MCC that significantly improves the quality of 3D object reconstruction. The proposed method employs a set of anchor points to provide global coarse structure prediction and utilizes the Neighborhood decoder to accelerate inference speed while enabling the exploitation of finer-scale visual features for improved recovery of 3D textures. Additionally, the Repulsive Unsigned Distance Function (UDF) is utilized to replace the occupancy field for more accurate surface representation, outperforming the traditional UDF in this problem.

The effectiveness of the proposed NU-MCC method is demonstrated through experimentation on multiple datasets. Specifically, the method quantitatively outperforms MCC by 9.7% in terms of F1-score on the CO3D-v2 dataset. Moreover, it showcases its superior performance qualitatively in zero-shot settings such as iPhone RGB-D capture, AI-generated images, and ImageNet.

Overall, this paper provides valuable discussion into the challenges associated with 3D object reconstruction and demonstrates the efficacy of the proposed NU-MCC method in overcoming these challenges.


**Weaknesses:**

Overall, I find the method to be promising, but there are several concerns that need to be addressed regarding the evaluation section.

Firstly, I believe the analysis of the effects of different anchor point numbers is missing. Given that the use of anchor points is one of the key designs of the method, it is important to understand their effect on the performance of the model. Therefore, it would be beneficial to conduct experiments analyzing the effects of different anchor point numbers to improve the evaluation section.

Secondly, while the repulsive UDF is another important design of the method, there are no quantitative results provided to ablate its influence. Although a single visual comparison is presented in Fig. 5, more experimental results are needed to justify this design and its impact on the overall performance of the model.

Thirdly, there seems to be some inconsistency with regards to the resolution used for training and testing. While the supplementary material states that the model trained with 128128 image resolution can be directly applied to 256256 image inputs, the main paper suggests that the model is trained with a 256256 resolution. It would be better to clarify these discrepancies and consider using a higher resolution such as 512512 for testing.

Furthermore, no video results are shown to check the 3D reconstruction results, which could provide a more comprehensive evaluation of the proposed method.

Lastly, there is a need to explain the design philosophy behind the aggregation weight estimation in Equation 2. It is unclear why the sum operation is used when the input to the network is the sum of three features. Therefore, providing more insights into the selection of weights and the reasoning behind it would enhance the clarity of the method.

In conclusion, while the method shows promise, the issues outlined above need to be addressed to improve the evaluation section and to strengthen the overall contribution of the paper.


**Questions:**

- Analysis of using different numbers of anchor points.
- More results (both quantitative and qualitative) to justify the repulsive UDF representation.
- Higher image resolution for testing.
- Video results for the reconstruction.
- Design of aggregation weight estimation.
More details can be found in the Weaknesses part.


**Limitations:**

Yes.

---

> ### Author Rebuttal · Authors · 2023-08-10
>
> We thank the reviewer for recognizing the value and promise of our method. We address your concerns as follows.
>
> **Analysis of the effect of different anchor point numbers:** To study this effect, we retrain our model with different numbers of anchor points. Given the constrained timeframe of the rebuttal phase, we conduct training on the “plant” object class. Results on plant validation set are as follows:
>
> | **n anchors**     | **Color (&#8595;)** | **Prec (&#8593;)** | **Recall (&#8593;)** | **F1 (&#8593;)** | **Acc (&#8595;)** | **Comp (&#8595;)** | **CD (&#8595;)** |
> |-------------------|-----------|----------|------------|--------|---------|----------|--------|
> | **10**            |     0.348 |     81.6 |       75.4 |   78.1 |   0.110 |    0.152 |  0.262 |
> | **50**            |     0.333 |     79.8 |       77.8 |   78.5 |   0.114 |    0.144 |  0.258 |
> | **100**           |     0.330 |     82.7 |       76.6 |   79.2 |   0.105 |    0.148 |  0.253 |
> | **200 (default)** |     0.330 |     81.4 |       77.7 |   79.2 |   0.108 |    0.142 |  0.250 |
>
> It is interesting to see that as few as 10 anchor points are able to well represent the overall structure of the objects. While a higher number of anchor points typically improves the reconstruction quality, the gain becomes smaller with more anchors.
>
> **Quantitative results of Repulsive UDF:** The quantitative comparison between the Repulsive UDF and the standard UDF is provided in the last two rows of Table 1 of our paper. More qualitative comparisons are provided in Figure A of the PDF.
>
> **Input resolution:** To clarify, there are two types of input resolutions in our pipeline. The first one is the input resolutions to be passed to the encoder ViTs. These resolutions are kept fixed during training and inference (224 x 224 for image and 112 x 112 for depth).
>
> The second type is the resolution for the mapping of fine features (seen points) in the feature aggregation process. We use a resolution of 112 x 112 during training and all of our evaluations (e.g., Table 1). Since this mapping process is not restricted to any network structure, we can map into arbitrary resolution during inference. As shown in the supplement, we can leverage this flexibility to map higher resolution (e.g., 224 x 224) seen points to recover finer details. However, using finer resolution incurs additional computational cost since every query point would then need to search more seen points to find the closest distances. Employing very high resolution (e.g., 448 x 448) therefore leads to computationally heavy inference, requiring the query points to search from 16x more points compared to the default resolution of 112 x 112.
>
> **Video results:** We provide video results at the link.
>
> **Design philosophy behind the aggregation weights:** As introduced on Line 158-161, we predict the aggregation weights based on three types of the information of the query point: 1) The displacement between $q$ and the anchors; 2) The anchor features; 3) The global token providing a global context. To fuse these information, we use the widely-used sum operation for its simplicity, and linear projection layers are used to adjust the features for better fusion. While there are other alternative operations, such as concatenation or multiplication, the specific choice is not a focus of this work, which we leave for future explorations.

---

> > ### Comment · Reviewer_WpNt · 2023-08-14
> > **Replying to Rebuttal by Authors**
> >
> > The authors' response resolve most of my concerns. I have upgraded my rating from weak accept to accept. Thanks.

---

### Official Review · Reviewer_97oK · 2023-07-01

**Soundness:** 3 good
**Presentation:** 3 good
**Contribution:** 3 good
**Rating:** 6
**Confidence:** 4

**Summary:**

This paper embarks on the task of 3D reconstruction using single-view RGB-D inputs. Building on Multiview compressive coding (MCC), it proposes two enhancements: 1) incorporating nearest neighbors in the decoder to minimize query time, and 2) the utilization of a repulsive unsigned distance function (UDF) as the output representation. The former also introduces an intermediate representation of anchor points for efficient NN query, while the latter replaces occupancy, delivering a higher quality of 3D reconstruction.

**Strengths:**

This paper effectively builds upon the latest research in Multiview Compressive Coding (MCC), addressing two main limitations encountered in the original study: 1) a relatively slow query speed and 2) the lack of high-fidelity details in the 3D reconstruction. In my opinion, this research is original, and I particularly like the idea of the repulsive unsigned distance function (UDF), and I believe it provides contributions to scholarly discourse in this field.

**Weaknesses:**

The manuscript could benefit from a more in-depth explanation of the anchor predictor's impact when it is used as an intermediate representation, particularly when neighborhood points are drawn from both the anchors and the initial point cloud. Augmenting the work with an additional ablation study could aid in comprehending its effectiveness and provide more depth to the reader's understanding.

In the supplementary, the quality of the scene-level reconstruction is acceptable, but it does not seem to be remarkable. It would be beneficial if the authors could identify and discuss the major barriers preventing high-fidelity scene reconstruction in their study.

**Questions:**

1. Line. 175, the dimension of F_f is (HW)x(dx3), are all points unprojected to the 3D space?
2. Line. 254, does the number of GT points affect the model performance?

**Limitations:**

Yes

---

> ### Author Rebuttal · Authors · 2023-08-10
>
> Thank you for the overall positive opinion to the contributions introduced in our paper. We address your concerns below.
>
> **Ablation study on neighborhood points:** We conduct an additional ablation study: aggregating different numbers of anchor points and seen points in the proposed neighborhood decoder. Results are as follows:
>
> | **n anchors** | **n seen points** | **Color (&#8595;)** | **Prec (&#8593;)** | **Recall (&#8593;)** | **F1 (&#8593;)** | **Acc (&#8595;)** | **Comp (&#8595;)** | **CD (&#8595;)** |
> |---------------|-------------------|---------------------|--------------------|----------------------|------------------|---------|----------|--------|
> | 4             | 0                 |               0.449 |               80.6 |                 58.0 |             66.3 |   0.103 |    0.213 |  0.316 |
> | 0             | 4                 |               0.393 |               41.3 |                 21.0 |             24.7 |   0.188 |    0.607 |  0.795 |
> | 4             | 4                 |               0.320 |               84.8 |                 83.7 |             83.8 |   0.103 |    0.134 |  0.237 |
> | 12            | 12                |               0.331 |               88.1 |                 79.7 |             83.0 |   0.089 |    0.160 |  0.249 |
>
> Comparing row 1-3, we find that the features from predicted anchors and seen points are complementary to each other, and combining both features leads to high-quality reconstruction results.
> In addition, increasing the number of aggregation points exhibits a smoothing effect (example shown in Sec. 2.2 of the supplement). For row 4, as smoothing operations typically decrease variance, the recovered point cloud concentrates more on high-confidence regions (high precision) and reduces coverage of the whole surface (low recall).
>
> **Scene-level reconstruction:** When applying our approach to scene reconstruction, we observe that the majority of anchor points are concentrated on prominent structures, such as walls, floors, and ceilings. Conversely, only a minority of anchors are situated on finer structures like chairs, sofas, and lamps. This distribution could potentially limit the fidelity of scene reconstruction. In future work, we intend to investigate methods to encourage anchor points to better distribute on intricate structures to address this limitation.
>
> **Are all points unprojected to the 3D space?**  We only unproject the pixels in the foreground to 3D points and set the points outside the foreground to a very high value so that they are not selected in the feature aggregation.
>
> **Does the number of GT points affect the model performance?** A higher number of GT points will likely result in a finer reconstruction. However, it will incur slower and more memory intensive training since each query point needs to search for its nearest GT point to obtain the GT UDF.

---

> > ### Comment · Reviewer_97oK · 2023-08-12
> >
> > Thanks. LGTM.

---

### Official Review · Reviewer_yY6d · 2023-07-13

**Soundness:** 3 good
**Presentation:** 3 good
**Contribution:** 3 good
**Rating:** 6
**Confidence:** 4

**Summary:**

The paper proposes a novel framework for generating 3D object from single view RGB-D input. Based on the previous work MCC, this work implements a neighborhood decoder that is more efficient and a repulsive udf representation that has higher quality than the previous occupancy field.

**Strengths:**

The major improvements are sound. I appreciate the neighborhood decoder that efficiently combines coarse and fine-scale features for 3D generation. The repulsive UDF is also convincing.

**Weaknesses:**

- What limits the generation resolution? Is it because of the point cloud quality in Co3D? Does the framework support rendering arbitrary resolution result images at inference time?
- When inferencing on in-the-wild images, does the model require segmenting out the foreground mask? Fig.2 seems a bit confusing because the image involves a background.
- How to determine the focal length and depth scale for in-the-wild images?
- How are the camera poses set for the input at training and inference time? Do you use the ground truth camera poses provided in CO3Dv2 dataset? Are the generated 3d objects placed in their canonical pose, or are they aligned with the input camera (meaning that there's one-to-one mapping between input image and output 3d object?
-  Does the pipeline generate plausible 3d objects when the input depth is problematic (e.g. distorted / noisy / scaled)? For example, when working with monocular depth.
- How many neighbors are used for the feature aggregation module? What's the value of $m$ in Eq. (1)?

**Questions:**

Please refer to weakness.

**Limitations:**

Yes.

---

> ### Author Rebuttal · Authors · 2023-08-09
>
> Thank you for the appreciation of the novelties introduced in our paper. We address your questions below.
>
> **Generation resolution:** Since the proposed NU-MCC learns a continuous neural field (UDF), we can query an arbitrary number of query points at inference. Note that the point generation process is much more efficient than the baseline algorithm MCC: the proposed method has linear complexity w.r.t. number of query points in a small feature aggregation network, while MCC incurs quadratic complexity in its heavy Transformer decoder.
>
> **Mask:** Following MCC, we apply a foreground mask to the input depth map, which is obtained via an off-the-shelf segmentation model.
>
> **Focal length and depth-scale:** For in-the-wild images whose focal length is unknown, we manually tune the focal length to ensure reasonable (not overly stretched/squeezed/distorted) point clouds from the depth map. This tuning process is independent from the subsequent 3D reconstruction approach, and we apply the same focal length to both our method and the baseline approaches. Note that it is possible to automate this procedure by techniques such as LeReS (CVPR 2021), which will be explored in future work. For the depth scale, the unprojected point coordinates are normalized as in MCC.
>
> **Camera pose:** We use the ground truth pose from CO3D-v2 dataset. The 3D generation is aligned with the input camera and thus also with the unprojected seen points.
>
> **Generation with problematic depth:** 1) Noisy. Our model has been trained with depth images obtained from COLMAP that are noisy. Therefore, it will be robust to noisy depth maps up to a certain extent. Under more severe noisy depths, the proposed Neighborhood decoder allows us to aggregate more point features to alleviate the artifacts caused by the noise (as explained in the supplement Sec. 2.2). 2) Distorted. Distorted depths will lead to distorted seen points, which in turn will result in distorted reconstruction. 3) Scaled. The depth scale does not matter since the unprojected points will be normalized.
>
> **Number of neighbors:** As stated in Line 267, each query point attends to its 4-nearest coarse anchor features and 4-nearest fine features in the feature aggregation, i.e., $m=n=4$.

---

### Author Rebuttal · Authors · 2023-08-10

First and foremost, we thank all the reviewers for the time and effort in reviewing our paper. We believe all of the reviewers’ comments and suggestions provide useful feedback to improve the quality of our paper. We hope to be able to clarify some concerns and doubts in this rebuttal and discussion period. To address some of the reviewers’ comments, we include a PDF and video (link is shared to Area Chair).

---

### Comment · Area_Chair_dtPa · 2023-08-20
**Thanks for the rebuttal!**

Dear authors,

A quick note to express our gratitude for your timely rebuttal. Your clarifications, responses, and the video link greatly aid the review process.

Thank you,
Area Chairs

---

> ### Author Response · Authors · 2023-08-21
>
> Dear Area Chairs,
>
> Thank you for your thoughtful message and the recognition you've extended to us. We're also grateful to the reviewers for their constructive comments and valuable insights.
>
> Best Regards,
>
> Authors of Submission2013

---

### Decision · Program_Chairs · 2023-09-21

**Decision:**

Accept (poster)

**Comment:**

In this paper, the authors propose NU-MCC for single image 3D reconstruction. The paper improves MCC in two ways: 1) a "neighborhood decoder" for more efficient computation, and 2) a repulsive UDF for improving reconstruction quality.

After rebuttal, all reviewers find this paper making solid contribution to our field and recommend acceptance. The AC agrees with the decision.